# New Antimicrobials for Gram-Positive Sustained Infections: A Comprehensive Guide for Clinicians

**DOI:** 10.3390/ph16091304

**Published:** 2023-09-15

**Authors:** Davide Carcione, Jari Intra, Lilia Andriani, Floriana Campanile, Floriana Gona, Silvia Carletti, Nicasio Mancini, Gioconda Brigante, Dario Cattaneo, Sara Baldelli, Mattia Chisari, Alessandra Piccirilli, Stefano Di Bella, Luigi Principe

**Affiliations:** 1Laboratory of Medicine and Microbiology, Busto Arsizio Hospital—ASST Valle Olona, 21052 Busto Arsizio, VA, Italy; davide.carcione@asst-valleolona.it (D.C.); gioconda.brigante@asst-valleolona.it (G.B.); 2Clinical Chemistry Laboratory, Fondazione IRCCS San Gerardo Dei Tintori, 20900 Monza, MB, Italy; jari.itra@irccs-sangerardo.it; 3Clinical Pathology and Microbiology Unit, Hospital of Sondrio, 23100 Sondrio, Italy; lilia.andriani@asst-val.it; 4Department of Biomedical and Biotechnological Sciences, Section of Microbiology, University of Catania, 95123 Catania, Italy; f.campanile@unict.it; 5Laboratory of Microbiology and Virology, IRCCS San Raffaele Scientific Institute, 20132 Milan, Italy; gona.floriana@hsr.it (F.G.); carletti.silvia@hsr.it (S.C.); 6Laboratory of Medical Microbiology and Virology, Department of Medicine and Technological Innovation, University of Insubria, 21100 Varese, Italy; nicasio.mancini@asst-settelaghi.it; 7Laboratory of Medical Microbiology and Virology, Fondazione Macchi University Hospital, 21100 Varese, Italy; 8Department of Infectious Diseases ASST Fatebenefratelli Sacco, 20157 Milan, Italy; dario.cattaneo@unimi.it; 9Pharmacology Laboratory, Clinical Chemistry Laboratory, Diagnostic Department, ASST Spedali Civili, 25123 Brescia, Italy; s.sarabaldelli@gmail.com; 10Microbiology and Virology Unit, Great Metropolitan Hospital “Bianchi-Melacrino-Morelli”, 89100 Reggio Calabria, Italy; mattia.chisari@ospedalerc.it; 11Department of Biotechnological and Applied Clinical Sciences, University of L’Aquila, 67100 L’Aquila, Italy; alessandra.piccirilli@univaq.it; 12Clinical Department of Medical, Surgical, and Health Sciences, Trieste University, 34129 Trieste, Italy; stefano932@gmail.com

**Keywords:** Gram-positive pathogens, novel anti-MRSA β-lactams, lipoglycopeptides, omadacycline, delafloxacin, tedizolid, PK/PD features

## Abstract

Antibiotic resistance is a public health problem with increasingly alarming data being reported. Gram-positive bacteria are among the protagonists of severe nosocomial and community infections. The objective of this review is to conduct an extensive examination of emerging treatments for Gram-positive infections including ceftobiprole, ceftaroline, dalbavancin, oritavancin, omadacycline, tedizolid, and delafloxacin. From a methodological standpoint, a comprehensive analysis on clinical trials, molecular structure, mechanism of action, microbiological targeting, clinical use, pharmacokinetic/pharmacodynamic features, and potential for therapeutic drug monitoring will be addressed. Each antibiotic paragraph is divided into specialized microbiological, clinical, and pharmacological sections, including detailed and appropriate tables. A better understanding of the latest promising advances in the field of therapeutic options could lead to the development of a better approach in managing antimicrobial therapy for multidrug-resistant Gram-positive pathogens, which increasingly needs to be better stratified and targeted.

## 1. Introduction

The emergence of antibiotic-resistant Gram-positive pathogens has made a shift in antibiotic usage policies necessary, prompting the exploration of novel molecules that can effectively overcome the spread of multiple mechanisms of resistance.

Clinical expectations from evidence on antibiotics against Gram-positive bacteria must be diversified according to their community-acquired or hospital-acquired nature; this includes the need for oral antibiotics that are effective against penicillin-resistant *Streptococcus pneumoniae* and methicillin-resistant *Staphylococcus aureus* (MRSA) for community-acquired infections such as respiratory and skin/soft-tissue infections. Otherwise, more effective antibiotics are needed for serious hospital-acquired infections such as bacteremia, ventilator-associated pneumonia, and endocarditis sustained by MRSA and vancomycin-resistant enterococci (VRE). The primary outcome of interest is all-cause mortality, with other outcomes including duration of hospital stay, resource use, adverse events, and resistance development.

In relation to the above-mentioned clinical expectations, some of the most promising novel antibiotics, including ceftobiprole, ceftaroline, dalbavancin, oritavancin, omadacycline, tedizolid, and delafloxacin will be examined. Based on existing evidence, this review provides an overview of the more recent studies on their molecular structure, mechanism of action, microbiological targeting, pharmacokinetic/pharmacodynamic properties, clinical use, and potential for therapeutic drug monitoring.

### A Global Overview of Genes Encoding Resistance to Anti-Gram-Positive Antibiotics

Resistance to anti-Gram-positive front-line antibiotics is a progressing phenomenon. Innumerable antimicrobial resistance (AMR) genes have been detected among the most clinically relevant species, often transferred by mobile genetic elements (MGEs) within genera and/or species, implying diverse genetic and biochemical resistance mechanisms. 

In order to provide a comprehensive understanding of the new molecules being reviewed, a brief overview of the well-established resistance mechanisms exhibited by the multidrug-resistant (MDR) Gram-positive pathogens of greatest clinical importance is essential. Among them, MRSA and VRE are the major recognized ones, but *Enterococcus* and *Staphylococcus* spp. isolates showing linezolid resistance or decreased susceptibility to the novel tetracycline derivatives are becoming increasingly common. The *mec*A or *mec*C genes marks MRSA, coding for an alternative penicillin-binding protein with low affinity for beta-lactams, which are responsible for high-level resistance to methicillin (and oxacillin, as a reference molecule) and other beta-lactam antibiotics in *S. aureus* [1,2,3]. VRE carry a plethora of *van* genes (informally defined *van*-alphabet), encoding for enzymes that modify the peptidoglycan conventional structure, preventing vancomycin binding to the D,Ala-D,Ala precursor; *van*A and *van*B are the most common in clinically important species (*Enterococcus faecium* and *Enterococcus faecalis*), conferring high-level resistance to vancomycin/teicoplanin, or only vancomycin, respectively [4]. The most reported *van*A gene was also rarely isolated to other co-infecting bacteria such as *S. aureus* [5].

High-level beta-lactam resistance in *E. faecium* isolates is characterized by the enhanced production of PBP5 and/or polymorphisms in its transpeptidase domain [6]; conversely, penicillin resistance, associated with reduced susceptibility to novel cephalosporins, has recently been found in *E. faecalis*, the usually less resistant enterococcal species, due to mutations in the *pbp*4 gene, which codes for a class B PBP4 with a low-affinity for beta-lactams [7].

A better understanding of the molecular mechanisms of resistance towards cell-wall damaging agents could play a crucial role in the quest for novel targets as antibacterial agents. Linezolid resistance currently accounts for the growing emergence of enterococci and diverse staphylococcal species. Besides the most common 23S rDNA gene-dosage dependent mechanism, the highly mobilizable *cfr*, *optr*A, and *poxt*A resistance genes have been identified among staphylococci, enterococci, and other Gram-positive isolates of human and animal origin, due to mutations in some or all ribosomal operons [8]. The *cfr* gene codes for a ribosomal methyltransferase conferring cross-resistance to phenicols, lincosamides, oxazolidinones, pleuromutilins, and streptogramin A; the *optr*A and *poxt*A genes code for ABC-F proteins with a resistance mechanism of ribosomal protection against oxazolidinones (also against the newer tedizolid) and phenicols; *poxt*A also confers resistance to tetracycline [9].

*Enterococcus* and *Staphylococcus* spp. strains have developed resistance against tetracycline and related antibiotics through the presence of specific tet^R^ genes by either utilizing efflux mechanisms (*tet*K and *tet*L) or conferring ribosomal protection (*tet*M, *tet*O, and *tet*S) [10].

Tigecycline (TGC) resistance was also rarely associated with loss-of-function mutations in the transcriptional repressor *mep*R, resulting in a derepression and overexpression of the MepA efflux pump belonging to the multidrug and toxic compound extrusion (MATE) family of multidrug efflux pumps [11]. 

Resistance to fluroquinolones has emerged with clinical use of these agents and is common in clinically relevant Gram-positive pathogens. It is increasingly due to mutations in genes encoding DNA gyrase and topoisomerase IV, as well as the acquisition of plasmids carrying resistance genes. These mutations occur in a specific domain of the enzymes and result in a reduced binding of the drug to the enzyme-DNA complex. Additionally, regulatory genes that control the expression of efflux pumps in bacterial membranes can also acquire resistance mutations. These efflux pumps have a broad range of substrates, including quinolones and other antimicrobials. Low-level resistance can be insidious, stemming from the overexpression of native efflux pumps and the emergence of plasmid-mediated resistance mechanisms, which can facilitate the selection of higher-level resistance and add a plasmid linkage to multidrug resistance [12].

Resistance to these front-line anti-Gram-positive antibiotics is currently covered at a therapeutic level by the introduction of these new antibiotics that have completed their clinical trials and now are commercially available: ceftaroline, ceftobiprole, dalbavancin, oritavancin, omadacycline, tedizolid, and delafloxacin.

## 2. β-Lactams: Fifth Generation Cephalosporins

### 2.1. Ceftaroline

#### 2.1.1. Chemical Structure and Mechanism of Action

The chemical structure of ceftaroline is depicted in Figure 1. Ceftaroline fosamil is an N-phosphono prodrug of the fifth generation cephalosporin derivative ceftaroline, presenting two amino side groups located at positions 3 and 7, respectively. It is administered for the treatment of adults with acute bacterial skin and skin structure infections. Ceftaroline fosamil is quickly hydrolysed to its active form, ceftaroline, via plasma phosphatases. Then, ceftaroline binds to penicillin-binding proteins (PBPs), particularly PBP 2A and PBPs 1, 2, and 3, which are located on the inner membrane of the bacterial cell wall, and inactivates them. PBPs are enzymes involved in the periplasmic and membrane steps of peptidoglycan biosynthesis, the main component of the bacterial cell wall. Therefore, the inactivation of PBPs affects the cross-linkage of peptidoglycan chains, thus weakening the bacterial cell wall and cell lysis [13].

#### 2.1.2. Microbiological Target

The spectrum of activity for ceftaroline includes major pathogens found in acute bacterial skin and skin structure infections (ABSSSI) and community-acquired bacterial pneumonia (CABP). Ceftaroline is a bactericidal antibiotic with a high affinity for specific penicillin-binding-proteins (PBPs) responsible for methicillin resistance in staphylococci (PBP2a) and non-sensitivity to penicillins in pneumococci (PBP 2x/2b) [14] (Table 1). It shows broad-spectrum activity against a range of Gram-positive bacteria, including methicillin-susceptible *S. aureus* (MSSA) and MRSA, vancomycin-intermediate *S. aureus* (VISA), heteroresistant VISA (h-VISA), vancomycin-resistant *S. aureus* (VRSA), daptomycin non-susceptible and linezolid-resistant *S. aureus*, *S. pyogenes*, *S. agalactiae*, and *S. pneumoniae* (including MDR strains), while its activity against *E. faecalis* is modest (not active against *E. faecium*) [15,16]. It shows activity for respiratory community-acquired Gram-negative pathogens such as *Haemophilus influenzae* and *M. catarrhalis*, including non-extended spectrum beta-lactamase producing Enterobacterales [17]. It has limited activity against most non-fermenting Gram-negative rods (i.e., *Pseudomonas aeruginosa*, *Acinetobacter* spp.) and many anaerobic species. Its activity against Gram-positive anaerobes, including *Peptostreptococcus* spp., *Propionibacterium* spp. and *Clostridium* spp., is similar to that of amoxicillin-clavulanate and 4–8 times that of ceftriaxone [18]. Development of resistance to ceftaroline occurs rarely in Gram-positive bacteria and at a similar rate to that of other oxyimino-cephalosporins in Gram-negative bacteria.

#### 2.1.3. Clinical Use

Ceftaroline has been approved by the EMA and FDA for the treatment of adults and children with CABP and ABSSSI. Pivotal clinical trials demonstrated non-inferiority to ceftriaxone in CAP and to aztreonam + vancomycin in cSSTI [19,20,21,22]. In real life, ceftaroline is an appealing option in patients with bacteremic MRSA pneumonia, with the advantages of hydrophily, good ELF penetration, and lower nephrotoxicity compared to vancomycin [23]. Its empirical use in monotherapy in nosocomial pneumonia is risky given its poor activity against Gram-negatives, especially Pseudomonas [24]. Ceftaroline has been successfully used as an add-on to daptomycin for persistent MRSA bacteremia [25,26]. Most off-label use is for these (in order): bacteremia, endocarditis, osteoarticular infections, HAP, and meningitis [27]. Animal studies showed a moderate CNS penetration with better antimicrobial activity compared to ceftriaxone + vancomycin in murine pneumococcal meningitis [28]. In human endocarditis and meningitis, ceftaroline is commonly used at higher doses (600 mg every 8 h) [27]. Encephalopathy [29] and neutropenia [30] are especially reported in patients with renal failure or with long therapies, respectively.

#### 2.1.4. PK/PD Characteristics

Ceftaroline is a fifth generation cephalosporin characterized by a linear PK profile following an IV infusion, with Cmax AUC values increasing in proportion to dose increases within the range of 50–1000 mg. The median steady-state volume of ceftaroline distribution is 20.3 L, and the average binding of ceftaroline to human plasma proteins is 20% (Table 2) [28,29,30]. Ceftaroline was found to be predominantly eliminated by glomerular filtration (90%). The mean half-life is 2.6 h in subjects with normal renal function, eventually increasing to 6 h in patients with ESRD [31,32,33]. Drug dosage adjustments have been established for patients with moderate to severe renal impairment. When compared with the PK parameters assessed in healthy younger adults aged 18–45 years, healthy elderly subjects (≥65 years) showed modest alterations, likely due to naturally occurring decreases in renal function over time. Chauzy et al. recently showed that ceftaroline clearance increased non-linearly in patients with augmented renal clearance (ARC), with a reduced probability of reaching PK/PD targets, especially in patients with MRSA infections [Therefore, higher than conventional doses should be considered in patients with ARC]. Studies in patients and healthy volunteers have reported good distributions of ceftaroline in the muscles (50%), subcutaneous tissues (50%), and in the epithelial lying fluid (22%); lower drug penetration was reported in the bone tissues (10%) and in CSF (6%) (reviewed in [34]). The PK/PD index that is associated with efficacy for ceftaroline is the percentage of time that free drug concentrations, usually measured in serum/plasma, are above the bacteria MIC during a dosing interval, (fT > MIC). When this index was assessed for ceftaroline in murine models, median fT > MIC values of 36% and 44% achieved a 1-log kill for *S. aureus* and *S. pneumoniae*, respectively.

#### 2.1.5. Potential Role of TDM

Liquid chromatography methods coupled with mass–mass spectrometry (LC-MS/MS) or with UV detection (HPLC-UV) have been validated for the TDM of ceftaroline [35,36]. The clinical PK/PD targets for cephalosporin efficacy are set at 45–100%fT > MIC, based on the severity of the infection (MRSA vs. MSSA) and on the patients’ conditions (more aggressive targets should be considered in ICU settings) [37]. No upper safety thresholds for ceftaroline C_min_ concentrations have been identified yet.

### 2.2. Ceftobiprole

#### 2.2.1. Chemical Structure and Mechanism of Action

The chemical structure of ceftobiprole is depicted in Figure 2. Ceftobiprole is a fifth generation cephalosporin belonging to thiadiazoles and presenting two amino side groups located at positions 3 and 7, respectively. It was developed for the treatment of hospital- and community-acquired pneumonia and complicated skin infections. Ceftobiprole shows activity by binding to PBPs, inhibiting the cross-linkage of peptidoglycan chains and the formation of the bacterial cell wall, leading to cell lysis and death. This drug can bind to different PBPs detected in both Gram-negative and Gram-positive bacteria. Ceftobiprole forms a stable acyl-enzyme complex with PBP2a and PBP 2x, and the combination with the long side chain located in the PBP 2′-binding pocket increases the stability of the bond and inhibition of the enzymes [38]. 

#### 2.2.2. Microbiological Target

Ceftobiprole is a fifth generation cephalosporin that has extended activity against a wide spectrum of Gram-negatives and Gram-positives. Concerning Gram-negatives, ceftobiprole retains activity against *Pseudomonas aeruginosa*, most members of the order *Enterobacterales*, and some anaerobic bacteria [39]. It maintains its stability against a wide variety of β-lactamases such as TEM and SHV types [40]. Ceftobiprole has major activity against MRSA, penicillin-resistant *S. pneumoniae*, and ampicillin-susceptible *E. faecalis*. In all cases, its activity is due to its high-affinity binding to the penicillin-binding proteins (PBPs), including the acquired PBP2a of MRSA strains, PBP2a of *Staphylococcus epidermidis*, and PBP2x of penicillin-resistant *S. pneumoniae*, resulting in the blocking of wall synthesis and bacterial death [41]. Ceftobiprole is degraded by acquired extended-spectrum β-lactamases (ESBLs) and carbapenemases (both serine-carbapenemases and metallo-carbapenemases), and it is also degraded by some class D enzymes. Ceftobiprole does not bind to PBP5; therefore, it is not active on *E. faecium*. The possibility of acquired resistance to ceftobiprole appears to be low; multiple genes and pathways may be potentially involved but are yet to be precisely defined [39,42].

#### 2.2.3. Clinical Use

Ceftobiprole has been approved by major European countries and several non-European countries (excluding the US) for CAP and HAP. Pivotal studies demonstrated non-inferiority to ceftriaxone (+optional linezolid if MRSA suspected) for CAP [43] and to ceftazidime + linezolid for HAP (while failing to reach non-inferiority for VAP) [44]. In 2021, a phase 3 trial demonstrated non-inferiority of ceftobiprole to vancomycin + aztreonam in ABSSSI [45]. From a clinical point of view, the potential of ceftobiprole is similar to that of ceftaroline, where ceftaroline has lower MICs for *Staphylococcus* while ceftobiprole has lower MICs for *Enterobacterales* with a degree of stability to AmpC enzymes but not ESBL [46]. In real life, ceftobiprole is commonly used for endocarditis (42%), bone and joint infection (24%), and HAP (15%), proving that most of its use is off label [47]. Good cure rates in patients with endocarditis, who were treated with daptomycin and ceftobiprole in combination, have also been reported [48]. From a safety point of view, ceftobiprole is expected to be a weak inducer of a *Clostridioides difficile* infection compared to other cephalosporins [49]. Its safety profile appears good, but data still need to be more conclusive. 

#### 2.2.4. PK/PD Characteristics

Ceftobiprole is a fifth generation cephalosporin administered via IV as a prodrug (ceftobiprole medocaril) and rapidly converted by plasma esterases into its active form. This cephalosporin presents linear PK, with an elimination half-life of about 3 h, a low protein binding (16%), and a volume of distribution of 18–20 L [50,51,52]. Ceftobiprole has a limited metabolism, and its major route of elimination is through glomerular filtration. Accordingly, drug clearance is reduced in patients with impaired renal function, and drug dose reduction based on creatinine clearance has been established in this clinical setting [52]. Conversely, prolonged drug infusions have been suggested for patients with ARC to optimize ceftobiprole exposure [52]. No differences in ceftobiprole PK were found in patients with or without ECMO [53].

Studies in healthy volunteers have reported a good penetration of ceftobiprole in the muscles (69%), adipose tissues (49%) and in epithelial lying fluid (26%); lower drug penetration was reported in bone tissues (22%) [54,55,56,57]. Similar to other cephalosporins, ceftobiprole shows time-dependent antibacterial activity. A recent population pharmacokinetic/pharmacodynamic analysis demonstrated that at the standard ceftobiprole dose and at a pharmacodynamic target of 60% t > MIC of the dosing interval, more than 90% of the population was adequately exposed to the drug [58].

#### 2.2.5. Potential Role of TDM

LC-MS/MS and HPLC-UV methods have been validated for the TDM of ceftobiprole [58,59,60,61]. In a case report, the clinical PK/PD target for ceftobiprole efficacy was set at 100%fT > 4xMIC for the treatment of an ICU patient undergoing continuous renal replacement therapy [62].

## 3. Lipoglycopeptides

### 3.1. Oritavancin

#### 3.1.1. Chemical Structure and Mechanism of Action 

The chemical structure of oritavancin is depicted in Figure 3. Oritavancin is a semisynthetic glycopeptide, structurally correlated to vancomycin, and it possesses the highly hydrophobic N-alkyl-p-chlorophenylbenzyl group and two 4-epi-vancosamine residues. Oritavancin was initially approved by the FDA in 2014 for the treatment of acute bacterial skin and skin structure infections. Oritavancin is capable of binding to the d-alanyl-d-alanine peptidoglycan termini and inhibits transpeptidation, which is the other essential enzymatic step in peptidoglycan polymerization. When oritavancin binds to newly formed template peptidoglycan chains near the cell membrane, the bacterial transpeptidase is blocked by steric interference associated with the hydrophobic tail group of oritavancin. Therefore, the inhibition of transpeptidase causes the cross-linking of adjacent peptidoglycan chains, and cell wall integrity and cell survival are compromised [63]. 

#### 3.1.2. Microbiological Target

Oritavancin, LY333328, is a semisynthetic glycopeptide derived from the N-alkylation of (LY264826), a naturally occurring glycopeptide with the same core structure as vancomycin [64,65]. Oritavancin exerts its activity through different mechanisms of action; like the glycopeptides vancomycin and teicoplanin, it inhibits cell wall synthesis by inhibiting the transglycosylation (polymerization) and transpeptidation (crosslinking) steps [66]. Additionally, oritavancin differs from vancomycin and teicoplanin by inhibiting bacterial RNA synthesis [67], collapsing transmembrane electrochemical potential, and increasing membrane permeability [68]. It was approved in the United States in 2014 and in Europe in 2015 for the treatment of acute bacterial infections of the skin and skin structures in adults [69]. Studies have demonstrated potent in vitro activity of oritavancin against MSSA (MIC_90_ 0.06 mg/L), MRSA (MIC_90_ 0.06 mg/L), as well as heterogeneous vancomycin-intermediate *Staphylococcus aureus* (hVISA) and VISA isolates and vancomycin-susceptible, methicillin-susceptible coagulase-negative *Staphylococcus* (MS-CoNS) (MIC_90_ 0.06 mg/L) and methicillin-resistant coagulase-negative *Staphylococcus* (MR-CoNS) (MIC_90_ 0.06 mg/L), [70]. Oritavancin exhibits potent activity against Enterococci, including vancomycin-resistant (VRE) variants with VanA and VanB phenotypes (MIC_90_ 0.06 mg/L); in summary, unlike other glycopeptides, oritavancin is able to bind to depsipeptides, including D-alanyl-D-lactate residues, which are present in organisms exhibiting VanA type resistance [71]. In addition, oritavancin has activity against other Gram-positive bacteria such as β-hemolytic Streptococci (BHS) (MIC_90_ 0.12 mg/L) and viridans group Streptococci (VGS) (MIC_90_ 0.06 mg/L). Gram-negative bacilli are intrinsically resistant to oritavancin and other lipoglycopeptides because the presence of the outer membrane prevents these molecules from entering the cell and binding to their target sites. Oritavancin resistance among clinical isolates has not been detected yet; non-susceptible isolates are rare or not yet reported [72]. However, in the laboratory, what has been observed among *Enterococcus* isolates showing VanA and VanB phenotypes is a moderate level of resistance to oritavancin by various mechanisms, such as the total replacement of the peptidoglycan precursors terminating D-alanine in D-lactate, the expression of the *van*Z gene, and the mutation in the *van*SB sensor gene of the Van-B cluster, which confer cross-resistance to teicoplanin and oritavancin [73].

#### 3.1.3. Clinical Use

Oritavancin has been approved by the EMA and FDA for ABSSSI. Its long half-life is well suited for use in the emergency department, especially in ABSSSI patients with poor compliance. However, the clinical potential of this drug is much broader, and its off-label administration is increasingly used. Oritavancin’s multimodal mechanism of action alongside its solid antimicrobial activity and high lipophilicity makes it an interesting option for parenchymal infections and bloodstream infections where low MICs can compensate for a theoretically suboptimal PK. Indeed, case series on its use in BSI are increasingly reported [74]. An interesting synergism has been reported between oritavancin and fosfomycin, thus combining a lipophilic and a hydrophilic drug [75]. The PK/PD of the drug will likely push its use in osteomyelitis. Real-life data on oritavancin (single or multiple dose) for osteomyelitis reported a clinical cure of 65% [76]. From a safety perspective, although oritavancin is a “lipophilic vancomycin”, data report ~30% fewer adverse events than vancomycin [77]. Use of intravenous unfractionated heparin sodium is contraindicated for 5 days after the administration of oritavancin [78].

#### 3.1.4. PK/PD Characteristics

Oritavancin is a novel lipoglycopeptide characterized by a long half-life (250 h), a high binding to plasma protein (85%), and an extensive tissue distribution (volume of distribution of 80 L) (Table 2) [63,74,79]. Dose-ranging studies have shown that this antibiotic displays linear pharmacokinetics, with renal excretion being the primary route of elimination [63,74,79]. No dosing adjustments are recommended in obese patients or those with moderate renal or hepatic impairment. No data are available on the PK of oritavancin in patients with severe renal insufficiency, in those undergoing intermittent or continuous renal replacement therapies, on ECMO, as well as in those with ARC. Although the drug is approved as a single dose for the treatment of ABSSSIs, there is increasing interest in the on-label use of oritavancin for complicated Gram-positive infections as an alternative to in-hospital intravenous or outpatient antimicrobial therapy [80]. Indeed, in an experimental animal model, it has been shown that, after a single injection, oritavancin is rapidly distributed to bone tissues, with concentrations remaining stable for up to 168 h, and bone tissue-to-plasma ratios ranging from 110 to 310% [81]. By performing a population PK analysis, Rose et al., documented that a simulated oritavancin regimen of a 1200 mg dose followed 7 days later by an 800 mg dose achieved drug concentrations above the susceptibility breakpoint (0.12 mg/L) for up to 8 weeks, and maintained a high AUC/MIC ratio for efficacy against organisms with MICs up to 0.25 mg/L [82]. This evidence, although preliminary, provides a strong rationale for the use of oritavancin for the treatment of osteoarticular infections. Drug–drug interaction studies reported that oritavancin is a weak inhibitor of CYP2C9/2C19 and a weak inducer of CYP3A4 and 2D6 [1,2,3,12]. The clinical relevance of these effects is, however, presently ill defined. Oritavancin can significantly interfere with the TDM of vancomycin and with some coagulation tests, leading to false positive/negative results [83,84].

#### 3.1.5. The Potential Role of TDM

To date, no analytical methods have been published for the measurement of oritavancin concentrations in biological matrices. No safety/efficacy PK/PD thresholds have been identified yet for the optimization of oritavancin use in clinical practice by TDM.

### 3.2. Dalbavancin

#### 3.2.1. Chemical Structure and Mechanism of Action

The chemical structure of dalbavancin is depicted in Figure 4. Dalbavancin is a second generation semi-synthetic lipoglycopeptide antibiotic, presenting bactericidal activity against a variety of Gram-positive bacteria. It contains a long lipophilic side chain that increases its potency and half-life. Upon administration, Dalbavancin tightly binds to the D-ala-D-ala portion of peptidoglycan chains, a site different from that of penicillins and cephalosporins, preventing peptidoglycan elongation and interfering with bacterial cell wall synthesis. This fact leads to bacterial cell death [85].

#### 3.2.2. Microbiological Target

Dalbavancin is a lipoglycopeptide antibiotic with bactericidal activity against Gram-positive bacteria, also showing in vitro anti-biofilm activity against staphylococcal biofilms. Gram-negatives are intrinsically resistant to DBV [86]. Dalbavancin shows broad-spectrum activity especially against Gram-positive cocci, with MIC_90_ values of 0.03 mg/L for S. aureus (also MRSA and MSSA, VISA and hVISA), β-haemolytic streptococci, viridans group streptococci, 0.06 mg/L for *E. faecalis* and *S. epidermidis*, and 0.12 mg/L for vancomycin-susceptible *E. faecium*. MICs determination should be performed in the presence of polysorbate-80 (0.002% P-80) in broth microdilution assays. It should be noted that dalbavancin, unlike oritavancin in the same family of lipoglycopeptides, does not cover the phenotype of vancomycin-resistant enterococci with a VanA resistance mechanism in its spectrum of activity [87,88]. Cross-resistance with other glycopeptides has been described, with an in vivo selection of dalbavancin non-susceptible VISA isolate [89]. 

#### 3.2.3. Clinical Use

Dalbavancin has been approved by the FDA and EMA for ABSSSI in 2014 and 2015, respectively. It was initially licensed as 1000 mg followed by 500 mg 1 week later, and was afterwards approved as 1500 mg in a single dose after demonstrating non-inferiority [90]. Dalbavancin is almost ten years old; thus, clinicians are quite familiar with its use. Apart from its in-label indications, an open-label trial for osteomyelitis (1500 mg IV on days 1 and 8) with promising results has been published, although the numbers are few [91]. A fair number of case series of endocarditis treated with dalbavancin have also been reported, with dalbavancin used as sequential treatment in most cases [92,93]. Encouraging retrospective results for BSI have also been published with sequential dalbavancin administered after a standard of care for at least 7 days for S. aureus bacteremia [94]. The safety profile is excellent [95]; however, in patients with known hypersensitivity to glycopeptides, caution is recommended.

#### 3.2.4. PK/PD Characteristics

Dalbavancin is characterized by a terminal half-life of >14 days, allowing administration either as a two-dose regimen (1000 mg day 1, 500 mg day 8) or as a single-dose regimen of 1500 mg for the treatment of ABSSSIs. The PK of dalbavancin in humans is linear, dose-proportional, and characterized by high protein binding [96]. After initial distribution, drug levels in plasma decline rapidly over the first 48 h as the drug distributes extensively into body tissues, including bone and articular tissue as well as epithelial lining fluid, with a total distribution volume of nearly 16 L (Table 2) [97]. Approximately one-third of the dose of dalbavancin is excreted in urine unmodified, with an additional one-third of dalbavancin excreted in feces and a further 12% excreted as a minor metabolite, hydroxyl-dalbavancin [98]. No dosage adjustment is necessary for patients with creatinine clearance > 30 mL/min, patients on hemodialysis, or those with mild hepatic impairment. No data are available on the PK of dalbavancin in patients undergoing continuous renal replacement therapies, on ECMO, as well as in those with ARC. Dalbavancin has also been used (off-label) in patients with osteoarticular infections. Dunne and coworkers have shown that, despite its long half-life, dalbavancin did not accumulate when given at the weekly dose of 500 mg up to 8 weeks [99]. By Monte-Carlo simulations, Cojutti et al. estimated that a two 1500 mg dosing regimen of dalbavancin 1 week apart may ensure efficacy against both MSSA and MRSA for up to 5 weeks [100]. 

#### 3.2.5. Potential Role of TDM

LC-MS/MS HPLC-UV methods have been developed and validated for the TDM of dalbavancin in routine clinical practice [101,102,103]. By performing TDM of dalbavancin used in a small series of ICU patients for the treatment of severe necrotizing fasciitis, we described a patient with a 5-fold increase in drug systemic clearance, probably related to the presence of severe hypoalbuminemia and continuous renal replacement therapy—two conditions that might have favored the increased dalbavancin renal elimination [104]. Subsequently, pillar works by Pea et al. provided preliminary evidence that the maintenance of total dalbavancin concentrations ≥4 or ≥8 mg/L (based on the infecting pathogen) over time could represent dalbavancin PK/PD efficacy thresholds [105,106]. No PK safety thresholds have been identified yet.

## 4. Tetracyclines

### 4.1. Omadacycline

#### 4.1.1. Chemical Structure and Mechanism of Action

The chemical structure of omadacycline is depicted in Figure 5. Omadacycline is the first aminomethylcycline, a novel class of substituted minocyclines derived by chemical modifications in the D ring of the tetracycline core. The C-9 substitution with an aminomethyl group provided a molecule with increased bioavailability compared to the glycylamido substitution that is present in tigecycline and eravacycline [4]. Omadacycline is used to treat moderate-to-severe infections including pneumonia, urinary tract infections, skin diseases, and blood-borne infections in both hospital and community settings. Omadacycline presents inhibitory activity in bacterial protein synthesis, as tetracyclines do. Moreover, in comparison to tetracyclines, omadacycline ensures similar activity in the presence and absence of ribosomal protection proteins or tetracycline efflux pumps [107].

#### 4.1.2. Microbiological Target

Omadacycline is a broad-spectrum tetracycline with activity against Gram-positive and a range of Gram-negative pathogens. Among Gram-positives, it shows activity against staphylococci, enterococci, and pneumococci, including MRSA (MIC_90_ 0.25 mg/L), VRE (MIC_90_ of 0.25 mg/L for E. faecalis and of 0.12 mg/L for *E. faecium*), penicillin- or macrolide-resistant *S. pneumoniae* (MIC_90_ 0.12 mg/L), and β-hemolytic streptococci (MIC_90_ 0.25 μg/mL). It also retains activity against tetracycline-resistant Gram-positive bacteria [108,109]. Among Gram-positive anaerobes, omadacycline has antimicrobial activity against *Clostridioides difficile* (MIC_90_ 0.5 mg/L), *Clostridium perfringens* (MIC_90_ 16 mg/L), and anaerobic Gram-positive cocci (MIC_90_ 1 mg/L) [108,109]. Like other tetracyclines, omadacycline displays no notable activity against *Proteus* spp. (MIC90 ≥ 32 mg/L), *Providencia* spp. (MIC90 > 16 mg/L), *Morganella* spp. (MIC90 > 16 mg/L), or *Pseudomonas* spp. (MIC90 >16 mg/L) [110,111,112]. Unlike older tetracyclines, omadacycline is active against bacterial isolates that express tetracycline-specific efflux pumps and/or ribosomal protection resistance mechanisms [108]. The high-level expression of *tet*(M) may confer resistance to omadacycline [113].

#### 4.1.3. Clinical Use

Omadacycline has been approved by the FDA for CABP and ABSSSI in 2018. The company withdrew its European application in 2019. Omadacycline is unaffected by the major mechanism of tetracycline resistance. Pivotal studies demonstrated non-inferiority compared to moxifloxacin for CABP [114] and to linezolid for ABSSSI [115]. In terms of antibacterial spectrum, omadacycline is similar to tigecycline but has a lower protein binding and is more hydrophilic; an oral formulation is also available [116]. PK data demonstrated that omadacycline has AUC0-24 in ELF and plasma 3-fold higher than tigecycline [117] and this could contribute to understanding the poor performance of tigecycline in lung infections. Given the increasing antibiotic resistance trend of intracellular bacteria, the in vitro activity of omadacycline against the *Legionella* [118], *Mycoplasma* [119,120], *Chlamydia* [118] and *Rickettsia* [121,122] species deserves to be mentioned. Omadacycline is also active in vitro against rapidly growing mycobacteria (e.g., *Mycobacterium abscessus*, *M. fortuitum* and *M. chelonae*) [123] with emerging clinical data [124]. Nausea occurs in more than 15% of cases [125] while a *C. difficile* infection is an infrequent complication [126]. 

#### 4.1.4. PK/PD Characteristics

Omadacycline is available as IV and oral formulations, the latter being characterized by poor bioavailability (35%), which can be significantly decreased under well-fed conditions (especially when dairy is included in the meal) [127,128]. The terminal half-life for omadacycline is about 16 h and, in order to achieve therapeutic concentration by day 2, loading doses of omadacycline are used [129]. Omadacycline has low binding to plasma proteins (21%) and a very high volume of distribution (200 L), two characteristics that explain the wide tissue distribution, with an important intrapulmonary penetration [130]. Omadacycline is not metabolized by metabolic enzymes; it does not induce or inhibit major cytochrome P450 enzymes and/or drug membrane transporters. Accordingly, omadacycline has a very low risk of being the perpetrator or victim of DDIs [118,131]. No omadacycline dose adjustment is warranted in patients with hepatic or renal impairment [132,133]. Sex was the only demographic characteristic significantly associated with drug PK (the systemic clearance was 16% lower in females than in males) [134]. In a population-PK study by Lakota et al., the concentration profile of omadacycline was best described by a linear, three-compartment model with zero-order intravenous infusion or first-order oral administration [135]. As previously reported for other tetracyclines, the AUC/MIC ratio is the PK/PD parameters that best correlate with the antibacterial efficacy of omadacycline in experimental animal models [136,137].

#### 4.1.5. Potential Role of TDM

To date, no analytical methods have been published for the determination of omadacycline concentrations in biological matrices. No safety/efficacy PK/PD thresholds have yet been identified for its optimization in clinical practice by TDM.

## 5. Oxazolidinones

### 5.1. Tedizolid

#### 5.1.1. Chemical Structure and Mechanism of Action

The chemical structure of tedizolid is depicted in Figure 6. Tedizolid is a member of the class of pyridines that present a D-ring substituent and a hydroxymethyl group in place of acetamide. Both of these substitutions are important to its clinical activity against some resistant pathogens. The additional phosphate group of the prodrug increases its aqueous solubility and bioavailability. It is administered as a phosphate prodrug that is rapidly converted by endogenous phosphatases to tedizolid. It is used for the treatment of acute bacterial skin and skin structure infections. Tedizolid binds to the 23S ribosomal RNA of the 50S subunit, preventing the formation of the 70S initiation complex and thus inhibiting protein synthesis [138].

#### 5.1.2. Microbiological Target

Tedizolid belongs to the second generation of oxazolidinones and is also indicated for the treatment of skin infections [139]. Tedizolid is a protein synthesis inhibitor active against multidrug-resistant Gram-positives, including MRSA, penicillin-resistant streptococci, and VRE [140]. Of note, tedizolid also displays in vitro activity against *C. difficile*, *Nocardia*, and nontuberculous mycobacteria [141,142,143,144]. It is not active against Gram-negatives, due to their outer membrane that constitutes a powerful barrier that hinders the entry of the drug. Resistance to tedizolid is very rare, since it also retains activity against linezolid resistant Gram-positives harboring the *cfr* gene-encoded methylase enzyme [145,146]. Chromosomal mutations in domain V of rRNA or ribosomal L3 or L4 proteins have been reported to confer resistance to both linezolid and tedizolid [147,148]. Moreover, the presence of an ABC transporter codified by the *optrA* gene carried by plasmids in enterococci confers resistance to phenicols and oxazolidinones [149].

#### 5.1.3. Clinical Use

Tedizolid has been approved by the EMA and FDA for ABSSSI after studies revealed it was noninferior to linezolid [150,151]. The molecule resembles linezolid for antibacterial spectrum and PK/PK properties. It is, therefore, not surprising that both in-label and off-label uses are similar to linezolid. Tedizolid has lower MICs but has more than double the rate of protein binding compared to linezolid [152]. Pivotal trials showed less thrombocytopenia in the tedizolid arms; however, post-marketing studies showed that this difference, if any, is minimal [153]. Tedizolid would, in theory, be a good option for CNS infection, but studies are lacking. Other interesting off-label indications are osteoarticular infections and mycobacterial infections, especially pulmonary ones caused by *M. abscessus*, with fair outcomes reported [154,155]. Tedizolid indeed has a good pulmonary penetration with AUC0-24 in ELF 40-fold higher than plasma (while that of linezolid is 5-fold) [156]. Its higher price compared to linezolid (after patent expiry) has likely contributed to the limit of its use in clinical practice.

#### 5.1.4. PK/PD Characteristics

Tedizolid is a second generation oxazolidinone marketed as a phosphate, a prodrug that is rapidly and extensively converted by plasma esterases to active moiety. Tedizolid is available both for IV use and as a film-coated tablet with an oral bioavailability >90% [157,158,159]. Major differences in the PK compared with linezolid relate to higher binding to plasma proteins (80%), longer half-life, and a larger volume of distribution (Table 2). Tedizolid is primarily metabolized by the liver as an inactive sulphate conjugate (phase II reaction), with no metabolism by cytochrome P-450 enzymes. Nevertheless, some clinically relevant interactions have also been reported for tedizolid [157,158,159]. Less than 20% of the drug is excreted unchanged in the urine. Tedizolid bactericidal activity on VRE and MRSA is time dependent. Correlations are closest between fAUC24/MIC and the tedizolid PK/PD index against MRSA and VRE. To achieve 1 log10 kill, tedizolid fAUC24/MIC in neutropenic mouse models with a thigh infection with VRE and MRSA should be 14.2 and 138.5, respectively. The post-antibiotic effects of tedizolid against VRE and MRSA are 2.39 and 0.99 h, respectively [160].

#### 5.1.5. The Potential Role of TDM

LC-MS/MS methods are available in the literature to monitor tedizolid concentrations in clinical practice [161,162,163]. However, no safety and/or efficacy clinical cutoff for tedizolid trough concentrations has yet been identified, and a TDM of tedizolid is presently considered unnecessary [164]. Nevertheless, in a phase 1 study in healthy volunteers, Lodise et al. aimed at characterizing the hematological profile of tedizolid given at 200, 300, or 400 mg once daily for 21 days and found that the magnitude of platelet count decreases from baseline was influenced by plasma drug concentration [165]. In particular, patients with a platelet count decrease > 20% had tedizolid trough concentrations >0.55 mg/L. This might be, therefore, a preliminary temptative safety cutoff to be considered to limit/prevent the hematological toxicity of tedizolid in patients requiring prolonged treatment with this drug. 

## 6. Fluoroquinolones

### 6.1. Delafloxacin

#### 6.1.1. Chemical Structure and Mechanism of Action

The chemical structure of delafloxacin is depicted in Figure 7. Delafloxacin is a fourth generation fluoroquinolone antibiotic with a special chemical structure, which makes it a weak acid. It presents a heteroaromatic substitution, which offers a larger molecular surface, and a chlorine atom, which increases its efficacy against anaerobic bacteria and provides strong polarity. Its large molecular surface can boost antibacterial activity against strains that are resistant to commonly used fluoroquinolones. Current guidelines approved delafloxacin for the treatment of adults’ acute bacterial skin and skin-structure infections, as well as community-acquired bacterial pneumonia. Like other fluoroquinolones, delafloxacin is a nucleic acid synthesis inhibitor, and its targets are bacterial gyrase and topoisomerase IV enzymes [166].

#### 6.1.2. Microbiological Target

Delafloxacin is a novel, broad-spectrum fluoroquinolone with antimicrobial activity against resistant Gram-positive, Gram-negative, and anaerobic organisms [167]. Among Gram-positive *S. aureus* isolates (MSSA and MRSA), delafloxacin MIC50/90 values were 0.004/0.25 mg/L, respectively [168]. A 2017 study by McCurdy et al. evaluated the activity of delafloxacin against a global collection of levofloxacin non-susceptible *S. aureus* isolates (*n* = 687), MRSA and MSSA isolates. The MIC_90_ values of delafloxacin were all 0.25 mg/L [169]. Delafloxacin also exhibited in vitro activity against methicillin-susceptible and methillin-resistant coagulase-negative staphylococci CoNS, *S. agalactiae*, *S. anginosus group*, *S. dysgalactiae*, *S. mitis group*, *S. pneumoniae*, and *S. pyogenes* [170]. EUCAST-specified MIC susceptibility breakpoints are the following: *S. aureus* (Acute bacterial skin and skin structure infections) and (Community-acquired pneumonia)≤ 0.25 mg/L, ≤0.016 mg/L, respectively; *S. pyogenes*, *S. dysgalactiae*, *S. agalactiae*, and *S. anginosus* group ≤0.03 mg/L [171]. It also retains activity against Gram-negative bacteria; *Enterobacterales* such as *Escherichia coli* (500 strains) MIC50/90 0.03 mg/L and 4 mg/L, respectively; *Klebsiella pneumoniae* (389 strains) MIC50/90 0.06 mg/L and >4 mg/L, respectively; *Pseudomomas aeruginosa* (200 strains) MIC50/90 0.25 mg/L and >4 mg/L, respectively [169]; and the *Escherichia coli* EUCAST breakpoint is 0.125 mg/L [171].

Delafloxacin has also demonstrated activity against the following: *Neisseria gonorrhoeae* MIC50/90 0.06 mg/L and 0.125 mg/L, respectively [172]; and *Moraxella catarrhalis* (100 strains) MIC 50/90 0.008 mg/L and *H. influenzae* (200 strains) MIC 50/90 0.001 mg/L and 0.004 mg/L, respectively [173], with a EUCAST breakpoint of the latter being 0.004 mg/L [173,174]. The broad spectrum of activity of delafloxacin, including atypicals, are classified in the following way: *Chlamydophila pneumoniae* MIC _90_ 0.125 mg/L [175], *Legionella pneumophila* MIC _50/90_ 0.12 mg/L, and *Mycoplasma pneumoniae* MIC_50/90_ 0.5 mg/L [176].

#### 6.1.3. Clinical Use

Delafloxacin has been approved by the EMA and FDA for ABSSSI and CABP. Pivotal studies demonstrated its non-inferiority to vancomycin + aztreonam for ABSSSI [177] and to moxifloxacin for CABP [178]. Its antibacterial spectrum is similar to ciprofloxacin/levofloxacin with the addition of MRSA and anaerobes [179]. Compared to the aforementioned quinolones, delafloxacin has a more than double protein binding and a lower volume of distribution [177]. It displayed good bone penetration in animal studies [180]; therefore, an off-label use for osteomyelitis is expected. Moreover, good anti-biofilm activity [181] and potent in vitro activity against both MSSA and MRSA makes delafloxacin an appealing molecule for prosthetic joint infections with a few cases having already been reported [182]. Given the good in vitro susceptibility, delafloxacin may also be a good candidate for nontuberculous mycobacterial infections [183]. Its increased activity in acidic environments (2 to 32-fold reduction in MICs) [184] makes delafloxacin worthy of consideration for abscesses/empyemas. From a safety point of view, similar adverse effects of quinolones are expected. Trials showed that the incidence of diarrhea was higher in delafloxacin arms [185].

#### 6.1.4. PK/PD Characteristics

Delafloxacin is a novel fluoroquinolone available as IV and oral formulations; the latter possesses a bioavailability of nearly 60% (the dose needed to be increased from 300 mg to 450 mg BID in patients who switched from the IV to the oral formulation) [186,187]. The drug is highly bound to plasma proteins, with a volume distribution of 32 L and a terminal half-life of 12 h (Table 2). Delafloxacin demonstrated high penetration into the lung compartments, as epithelial lining fluid concentrations were substantially higher than the free drug in plasma [186,187]. Nearly 50% of the drug is eliminated in the urine (as parent drug and glucuronide metabolites). Accordingly, a reduced IV dosage (200 mg) is recommended in patients with eGFR < 30 mL/min to avoid the accumulation of cyclodextrin (an excipient in the formulation) in renal tissues. No dose adjustments based on renal function are required with the oral formulation. Delafloxacin should not be administered in patients with eGFR < 15 mL/min [180]. However, in an open-label, parallel-group, crossover study, Hoover et al. reported that, compared to subjects with normal renal function, the maximum exposure to delafloxacin was 13% and 33% higher for ESRD subjects given delafloxacin 1 h before and 1 h after hemodialysis, respectively [188]. Remarkably, delafloxacin was well tolerated in ESRD subjects, with diarrhea being the most reported treatment-emergent adverse event. No PK data are available in patients on CCRT, ECMO, or with ARC. No dose adjustments of delafloxacin are required in the presence of mild, moderate, and severe hepatic impairment. In experimental models that studied neutropenic murine lung infection, the efficacy of delafloxacin was best demonstrated by the fAUC/MIC ratio, being the necessary magnitude to reach a 1 log10 CFU decrease (31.8, 24.7, and 9.6 for *S. pneumoniae*, MRSA and *K. pneumoniae*, respectively) [189]. In a phase 1 study in healthy subjects, Paulson et al. reported that steady-state dosing of delafloxacin produced no significant changes in midazolam pharmacokinetics (a CYP3A probe) and concluded that no clinically relevant inhibition of cytochrome P450 enzymes occurred during delafloxacin administration [190,191]. The oral bioavailability of delafloxacin can be significantly reduced if taken concomitantly with drugs or products that contain multi-valent metal cations (aluminum, magnesium, iron, zinc, or calcium) [186,187].

#### 6.1.5. Potential Role of TDM

Liquid chromatography methods have been developed for PK analyses of delafloxacin [188,189,192]. However, no TDM studies have been published and no PK/PD safety/efficacy thresholds have been identified yet.

## 7. Conclusions

Most severe nosocomial and community infections are caused by Gram-positive bacteria, eventually involving *Enterococcus* spp. and *Staphylococcus* spp., with decreased susceptibility to available antibiotics [193]. The disposal of new molecules on the market provides hope for treating patients infected by these microorganisms. In particular, this study discusses ceftobiprole, ceftaroline, dalbavancin, oritavancin, omadacycline, tedizolid, and delafloxacin. 

Considering the microbiological targets, these new antibiotics show complete antimicrobial activity on MRSA and β-hemolytic streptococci. Regarding VRE (mostly VanA-producers), an important antimicrobial activity can be obtained from oritavancin, omadacycline, and tedizolid, while new cephalosporins, dalbavancin, and delafloxacin do not represent optimal therapeutic options. For the MRSE and *Streptococcus* viridans group, oritavancin, dalbavancin, omadacycline, tedizolid, and delafloxacin represent good therapeutic options, while no activity is expected for ceftobiprole. Finally, new therapeutic options for *S. pneumoniae* infections are represented by ceftaroline, omadacycline, tedizolid, and delafloxacin.

Resistance to new anti-Gram-positive antibiotics is currently sporadic. However, the antimicrobial activity (and clinical efficacy) could be preserved only through appropriate dosing, use, and careful monitoring of the emergence of AMR. This review suggests that a cautious and optimal antimicrobial stewardship, also considering combination therapy including old and new molecules, is strongly advisable in order to preserve last-resort antibiotics.

From a pharmacologic perspective, the rationale selection of the best therapeutic option requires a good knowledge of the penetration of antibiotics in the site of infection and optimal matching of this information with patients’, drugs’, and hosts’ characteristics. For instance, omadacycline and tedizolid are, among the novel antibiotics, those characterized by the highest lung penetration, with ELF concentrations 3- to 5-fold higher compared with those reached in the systemic circulation, making them optimal therapeutic options for the treatment of MRSA pneumonia at marketed doses. Conversely, ceftobiprole and ceftaroline, with an ELF/plasma ratio of around 20–25%, may require optimized dosing regimens [44,194,195]. On the same line, dalbavancin and oritavancin, given their optimal penetration in bone structures and their long half-life, are ideal candidates for the long-term treatment of osteoarticular or prosthetic joint infections [96,97,127,128]. 

Beyond the issue of drug penetration, the reviewed antibiotics are also characterized by significant differences in the main PK feature, which may deserve further attention and future investigations. Indeed, ceftaroline, ceftobiprole, oritavancin, and delafloxacin are mainly eliminated through the kidneys. Proper drug dose adjustments are available for patients with different degrees of renal function but are completely lacking for patients with ARC and/or in patients on ECMO. Similarly, the PK of dalbavancin, oritavancin, tedizolid, and delafloxacin is likely to change significantly in patients with severe hypoalbuminemia (i.e., serum albumin < 2 g/dL), but no data are available from the literature in this clinical context. In the same way, body fluid retention (as a result of edema in sepsis and trauma, pleural effusion, ascites, fluid therapy, etc.) might significantly increase the Vd of ceftaroline, ceftobiprole, dalbavancin, and delafloxacin, potentially resulting in antibiotic underexposure. The presence of clinical conditions potentially altering the PK of the reviewed antibiotics could be handled in clinical practice by TDM [37]. However, data on the TDM are presently available only for ceftaroline, ceftobiprole, and dalbavancin with preliminary but very promising results [34,59,101,102,103]. 

From a clinical point of view, almost all of the new anti-Gram-positive antibiotics have been approved for ABSSSI, and 3 out of 7 of them have been approved for CABP (ceftaroline, ceftobiprole, omadacycline, and delafloxacin). Among them, ceftobiprole has also been approved for HAP. Apart from these three main in-label indications (ABSSSI, CABP, and HAP), the reporting of several off-label uses is increasing and is noteworthy for the near future [196]. Ceftaroline has interesting data for CNS infections; ceftobiprole has good real-life data for endocarditis; dalbavancin has a clinical trial for osteomyelitis; oritavancin is expected to increase its off-label use for BSI and bone infections; omadacycline has interesting potential against intracellular bacteria and mycobacteria; tedizolid has fair outcomes reported for mycobacterial infections; and delafloxacin retains good potential for bone infections and abscesses.

In conclusion, in consideration of their spectrum of activity, PK/PD characteristics, low toxicity, and the monitoring possibility of TDM, the new antibiotics described here currently represent important treatment options for infections sustained by MDR or XDR Gram-positives.

## Figures and Tables

**Figure 1 pharmaceuticals-16-01304-f001:**
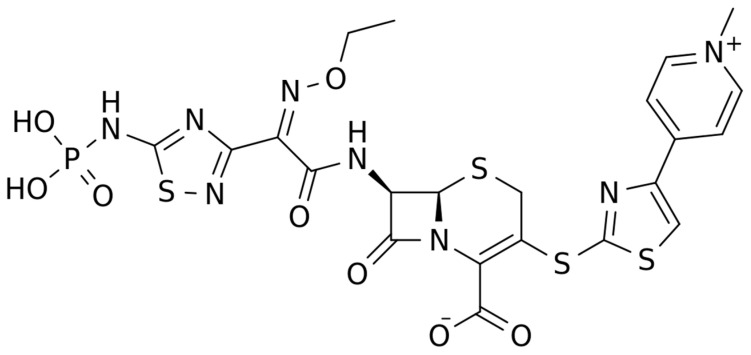
Chemical structure depiction of ceftaroline fosamil.

**Figure 2 pharmaceuticals-16-01304-f002:**
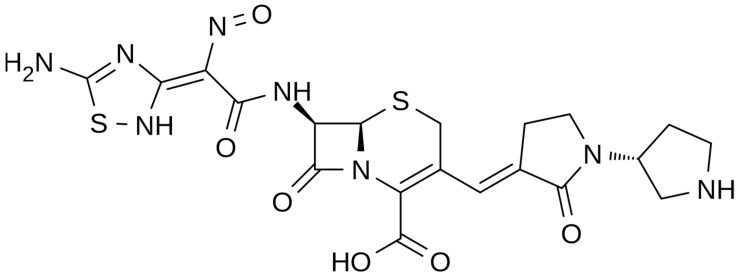
Depiction of the chemical structure of ceftobiprole.

**Figure 3 pharmaceuticals-16-01304-f003:**
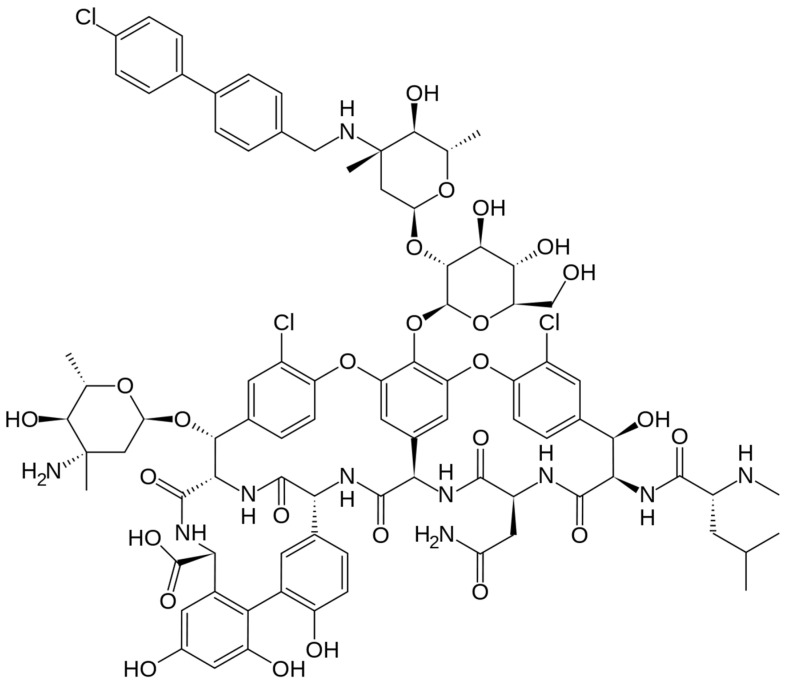
Depiction of the chemical structure of oritavancin.

**Figure 4 pharmaceuticals-16-01304-f004:**
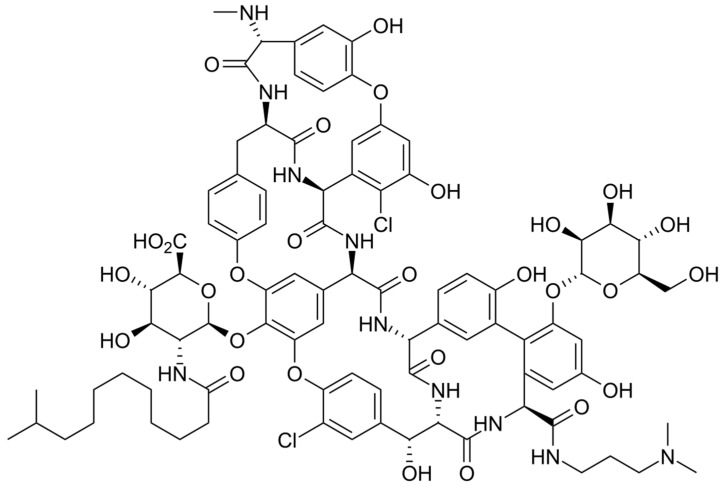
Depiction of the chemical structure of dalbavancin.

**Figure 5 pharmaceuticals-16-01304-f005:**
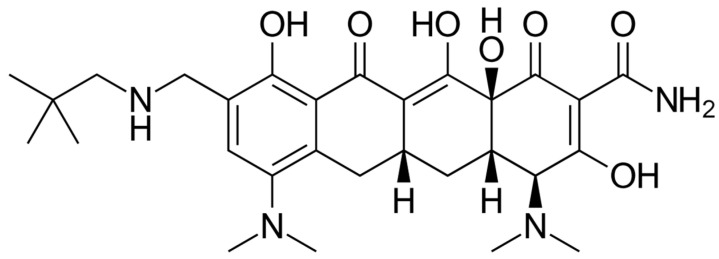
Depiction of the chemical structure of omadacycline.

**Figure 6 pharmaceuticals-16-01304-f006:**
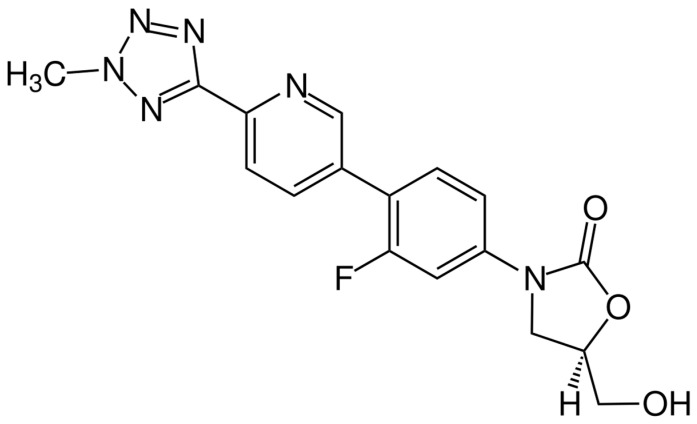
Depiction of the chemical structure of tedizolid.

**Figure 7 pharmaceuticals-16-01304-f007:**
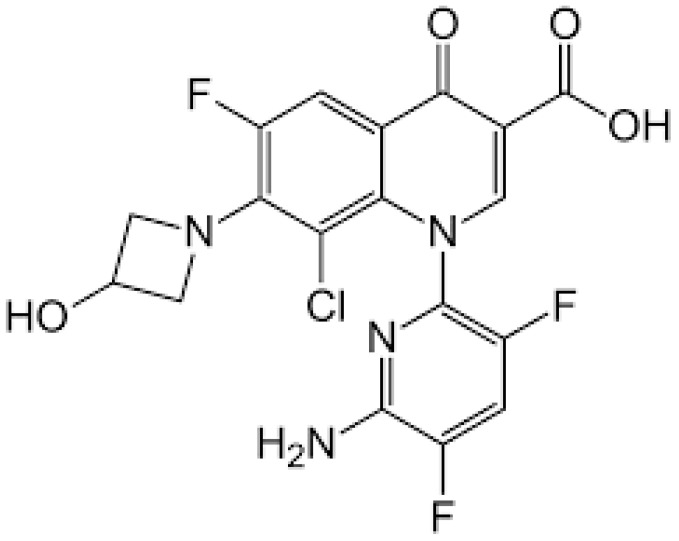
Depiction of the chemical structure of delafloxacin.

**Table 1 pharmaceuticals-16-01304-t001:** Microbiological targets, Green = antimicrobial activity, red = no antimicrobial activity, yellow = partial antimicrobial activity, MRSA: methicillin-resistant *S. aureus*; MRSE: methicillin-resistant *S. epidermidis*; VRE: vancomicin-resistant *Enterococcus* spp.

	MRSA	MRSE	VRE	Beta Hemolytic Streptococci	Viridans Group Streptococci	*Streptococcus pneumoniae*
**Ceftaroline**						
**Ceftobiprole**						
**Oritavancin**						
**Dalbavancin**						
**Omadacycline**						
**Tedizolid**						
**Delafloxacin**						

**Table 2 pharmaceuticals-16-01304-t002:** Summary of the main PK/PD characteristics of the novel antibiotics for the treatment of Gram-positive bacterial infections.

Drug	Half-Life h *	Vd L	Protein Binding %	Renal Elimination %	ELF/Plasma Ratio, % ^	CSF/Plasma Ratio, % ^	TDM in Routine	Clinical PK/PD Efficacy Target	Clinical PK Safety Target
**Ceftaroline**	2.6	20	20%	90%	22%	6%	Yes	45–100% fT > MIC	n.e.
**Ceftobiprole**	3.0	18	16%	90%	26%	2–16%	Yes	60–100% fT > MIC	n.e.
**Dalbavancin**	250	30	>90%	40%	36%	2%	Yes	Cmin > 4 (8) mg/L	n.e.
**Oritavancin**	300	80	85%	>90%	5%	1–5%	n.e.	n.e.	n.e.
**Omadacyclin**	16	200	21%	30%	147%	n.e.	n.e.	n.e	n.e.
**Tedizolid**	12	80	80%	<20%	300%	3–55%	n.e.	n.e.	Cmin < 0.55 mg/L
**Delafloxacin**	12	30	85%	50%	n.e	n.e.	n.e	n.e.	n.e.

Vd: volume of distribution, BP: protein binding, TDM: therapeutic drug monitoring; n.e.: not established; PK/PD: pharmacokinetic/pharmadynamic%fT > MIC: % of time between 2 administrations that free drug concentrations are above the minimum inhibitory concentration; Cmin: trough concentrations; ELF: epithelial lying fluid; CSF: cerebrospinal fluid. * in subjects with normal renal function; ^ based on data from experimental studies and/or clinical observations.

## Data Availability

Data sharing not applicable.

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
