# Peer review of "New Antimicrobials for Gram-Positive Sustained Infections: A Comprehensive Guide for Clinicians"

_pharmaceuticals, 2023, doi:10.3390/ph16091304_

Round 1
Reviewer 1 Report
The review article entitled “New antimicrobials for Gram-positive-sustained infections: chemical structure, mechanism of action, microbiological target, clinical use, PK/PD properties and TDM potential. A multidisciplinary management guide for clinicians” tries to provide the clinicians a guideline to manage the drug-resistant bacteria. Although some valuable information about each antibiotic has been gathered, I am wondering whether such studies need to be supported by systematic review and meta-analysis studies to be introduced as a guide or not. Please explain.
In addition, there are some important issues that need to be addressed.
#Title: While the current title effectively conveys the main focus of the review article, it could be improved by making it more concise and reader-friendly. For example, "New Antimicrobials for Gram-Positive Infections: A Comprehensive Guide for Clinicians"
#Abstrat needs substantial revision. The deficiencies in the abstract are Lack of specific objectives, Insufficient detail on methodology, Lack of context and significance, and Vague reference to provided tables.
#Keywords can be improved.
#Introduction:
Providing an introduction section is highly recommended for a review article. What has currently been titled as the introduction, does not meet the criteria of a sufficient introduction. The following issues can be identified in the given introduction; Improving these aspects will make the introduction more informative, coherent, and engaging for readers.
*Lack of clear objectives; *Insufficient contextual information (While the introduction mentions the progressing phenomenon of resistance to anti-Gram-positive antibiotics, it lacks specific background information about the topic); *Incomplete coverage of resistance mechanisms (The introduction briefly mentions some resistance mechanisms, such as the mecA gene in MRSA and van genes in VRE, but does not provide a comprehensive overview of the resistance mechanisms for Gram-positive bacteria); *Lack of transition or flow (The introduction abruptly jumps from discussing MRSA and VRE to mentioning resistance to linezolid, without a smooth transition or connection)
@The authors are suggested to use these articles to improve their introduction; https://doi.org/10.1186/s13568-021-01247-z , https://doi.org/10.3390/pharmaceutics14102049
Author Response
Reviewer 1
The review article entitled “New antimicrobials for Gram-positive-sustained infections: chemical
structure, mechanism of action, microbiological target, clinical use, PK/PD properties and TDM
potential. A multidisciplinary management guide for clinicians” tries to provide the clinicians a
guideline to manage the drug-resistant bacteria. Although some valuable information about each
antibiotic has been gathered, I am wondering whether such studies need to be supported by
systematic review and meta-analysis studies to be introduced as a guide or not. Please explain.
We thank the reviewer for this advice. Undoubtedly, a systematic review and meta-analysis studies
should be statistically more consistent but, in our opinion, they may not be necessary for the
purpose of this review, which would be to suggest new antimicrobials for Gram-positive-sustained
infections, in terms of chemical structure, mechanism of action, microbiological target, clinical use,
PK/PD properties and TDM potential. Unfortunately, most evidence on new anti-Gram-positive
antimicrobials comes from relatively limited data, often from not-randomized/not-controlled
studies. In situations like this, in our opinion, performing a systematic review or a meta-analysis
would make the manuscript cumbersome without adding a real methodological "plus". However,
we made ensured that the studies included in the review were reliable and of high quality to
provide accurate and useful information to clinicians.
In addition, there are some important issues that need to be addressed.
#Title: While the current title effectively conveys the main focus of the review article, it could be
improved by making it more concise and reader-friendly. For example, "New Antimicrobials for
Gram-Positive Infections: A Comprehensive Guide for Clinicians".
Thank you for this suggestion. The title has been changed accordingly.
#Abstrat needs substantial revision. The deficiencies in the abstract are Lack of specific objectives,
Insufficient detail on methodology, Lack of context and significance, and Vague reference to
provided tables.
According with the reviewer suggestions, the abstract has been substantially revised, and all the
implementations have been achieved.
#Keywords can be improved.
New Keywords have been included.
#Introduction:
Providing an introduction section is highly recommended for a review article. What has currently
been titled as the introduction, does not meet the criteria of a sufficient introduction. The
following issues can be identified in the given introduction; Improving these aspects will make the
introduction more informative, coherent, and engaging for readers.
*Lack of clear objectives; *Insufficient contextual information (While the introduction mentions
the progressing phenomenon of resistance to anti-Gram-positive antibiotics, it lacks specific
background information about the topic); *Incomplete coverage of resistance mechanisms (The
introduction briefly mentions some resistance mechanisms, such as the mecA gene in MRSA and
van genes in VRE, but does not provide a comprehensive overview of the resistance mechanisms
for Gram-positive bacteria); *Lack of transition or flow (The introduction abruptly jumps from
discussing MRSA and VRE to mentioning resistance to linezolid, without a smooth transition or
connection)
We agree with the Reviewer that the introduction section needed an extensive improvement.
According with the reviewers’ suggestion, a new revised version was provided, consistent with the
contents of the article. In this additional paragraph, the actual objectives and topics described in
the text, were highlighted.
We believe that understanding the role of main resistance mechanisms is crucial in the quest for
novel therapeutic options. Therefore, the overview of the major mechanisms that have led to the
spread of the most clinically relevant multidrug-resistant (MDR) Gram-positive pathogens has been
kept and converted into a separate paragraph, and improved accordingly to the reviewer
suggestions:
It has been extended with a part related to fluroquinolones resistance.
Linezolid and tetracycline resistance mechanisms was extensively improved, accordingly.
The part of this paragraph that lacked in fluency was changed, accordingly.
@The authors are suggested to use these articles to improve their introduction;
https://doi.org/10.1186/s13568-021-01247-z , https://doi.org/10.3390/pharmaceutics14102049A
As suggested by the reviewer, these articles were used in the revised version of the introduction.

Reviewer 2 Report
The manuscript received for evaluation is a review that deals with several antimicrobials commonly used; a description of the structures and properties is presented. A number of 198 references are found. Although this work can have some importance as a review that join together a large number of literature data, there is a feeling of simple compilation of data. Some issues are the following:
-a shorted title, up tp three lines is recommended;
-keywords should not be in bold;
-references numbers in text should not be in bold;
-Introduction is numbered as 1, followed by 1.1, but not 1.2 is found, therefore is no need to have 1.1;
-chemical structures should be bigger in size and without empty spaces between texts and figure, and not copied from the internet, there are plenty of free drawing chemical structure programs;
-tables are not in the recommended format of the journal;
-references are not in the correct format.
Based on these, a major revision is supposed to be done before publishing, taking into consideration the weakness as a simple report. Addition of similar data can also be considered. Figures and tables needs improvements. After these corrections, the manuscript can be accepted for publication.
Author Response
Reviewer 2
The manuscript received for evaluation is a review that deals with several antimicrobials
commonly used; a description of the structures and properties is presented. A number of 198
references are found. Although this work can have some importance as a review that join together
a large number of literature data, there is a feeling of simple compilation of data. Some issues are
the following:
-a shorted title, up tp three lines is recommended;
We agree with the reviewer. The title has been changed accordingly
-keywords should not be in bold;
The bold format has been removed and new keywords have been included
-references numbers in text should not be in bold;
The bold format has also been removed in the reference numbers throughout the text
-Introduction is numbered as 1, followed by 1.1, but not 1.2 is found, therefore is no need to have
1.1;
The introduction section was substantially revised and enhanced. An initial part has been added,
with the actual objectives and topics described in the text. However, a separate paragraph 1.1,
with a brief overview of the main resistance mechanisms that have led to the spread of the most
clinically relevant multidrug-resistant (MDR) Gram-positive pathogens, has been kept. We believe
that understanding the role of these mechanisms is crucial in the quest for novel therapeutic
options.
-chemical structures should be bigger in size and without empty spaces between texts and figure,
and not copied from the internet, there are plenty of free drawing chemical structure programs;
Figures of chemical structures have been changed
-tables are not in the recommended format of the journal;
Tables have been revised in the correct format
-references are not in the correct format.
References have been revised in the correct format
Based on these, a major revision is supposed to be done before publishing, taking into
consideration the weakness as a simple report. Addition of similar data can also be considered.
Figures and tables needs improvements. After these corrections, the manuscript can be accepted
for publication.
Thank you for all recommendations to improve the article. All corrections were done, according
with the reviewers’ suggestions. The introduction section was substantially revised; tables and
figures were improved, and all minor revision (bold, keywords, reference format) were done

Round 2
Reviewer 1 Report
The authors have effectively addressed all the pertinent issues and have provided satisfactory responses. Based on the thoroughness and quality of their work, I recommend their manuscript for publication.
Reviewer 2 Report
The authors took very seriously the suggestions made for improvement of the manuscript therefore I am glad to propose the acceptance of their work. Please check reference 202.